# Promoting or pressurising participation? A discourse analysis of online patient information resources about prehabilitation before cancer treatment

Sophie Stanley[1,2]*, Hilary Stewart[1], Cliff Shelton[1,3]

1 Lancaster Medical School, Lancaster University, Lancaster, United Kingdom, 2 North West School of Anaesthesia, Manchester, United Kingdom, 3 Department of Anaesthesia, Wythenshawe Hospital, Manchester, United Kingdom

* s.a.stanley@lancaster.ac.uk

## Abstract

Prehabilitation aims to improve outcomes by optimising health before treatment. Interventions typically target diet, physical activity and/or mental health. Communicating the benefits of prehabilitation may influence patients' engagement in interventions. However, the evidence for prehabilitation prior to cancer treatment is replete with uncertainties. Synthesising and communicating the efficacy of prehabilitation is challenging. This study aims to understand how evidence, motivation and accessibility are balanced in online patient-facing resources about prehabilitation. Databases, search engines and websites (identified by prehabilitation researchers) were systematically searched for patient-facing resources from UK organisations about prehabilitation before cancer treatment. Search strategies were built from non-technical synonyms for three terms: prehabilitation, cancer, and patient information. Results were screened against predefined eligibility criteria. The Quality Evaluation Scoring Tool assessment informed purposive sampling. Included resources were interrogated using discourse analysis. Screening of 3394 search results identified 68 resources from which a sample of 25 was analysed. Two themes summarised how resources presented prehabilitation to patients. Resources influenced rather than informed patients about participation in prehabilitation. Benefits were presented with emphasis, certainty and authority whereas limitations or alternatives were rarely discussed. The information focused on individual motivation rather than acknowledging patients' resources or systemic barriers. Overall, it functioned to convince patients to participate in prehabilitation. Promoting prehabilitation in patient-facing literature may be beneficial. However, this relies upon two assumptions: firstly, that this communication approach is effective at increasing participation in practice, and secondly, that prehabilitation itself is 'beneficial'. When outcomes prioritised by patients are not established, and evidence remains uncertain, this is not guaranteed. Overpromoting

**Data availability statement:** The data used in this study are already publicly available. We sought to protect the identities of the organisations who produced this data. We have chosen not to share openly the individual resources or websites analysed as this would compromise that aim. Further information about the data used and conditions for access are available from Lancaster University's Institutional Repository at doi.org/10.17635/lancaster/researchdata/143.

**Funding:** SS is an Academic Clinical Fellow whose salary is part funded by the National Institute for Health and Care Research (NIHR). The salary for HS was funded by the NIHR Health Services and Delivery Research programme as part of the PARITY study (NIHR134282). The views expressed are those of the authors and not necessarily those of the NIHR or the Department of Health and Social Care. The funder did not play any role in the study design, data collection and analysis, decision to publish, or preparation of the manuscript. There was no additional external funding received for this study.

**Competing interests:** The authors have declared that no competing interests exist.

the benefits of prehabilitation risks giving patients unrealistic expectations. Allocating responsibility to individuals may risk introducing patient blame and guilt in the event of treatment complications. Further research is required to understand how patients experience information resources and to define the patient-centred outcomes of prehabilitation.

## Introduction

Prehabilitation describes the concept of enhancing functional capacity before (and potentially during) treatment in order to improve post-intervention outcomes [1]. In general, it seeks to improve physical and psychological health through a series of interventions, most often targeting diet, physical activity and mental wellbeing. Prehabilitation programs are complex behavioural interventions which require significant engagement from patients [2].Models of behaviour change, referenced in current UK prehabilitation guidance, recognise motivation as key to changing behaviours [3,4]. In previous studies, patients have stated that they are motivated to participate when they understand the potential benefits of interventions [5–8]. Healthcare professionals need to communicate the benefits of prehabilitation effectively in order for it to be motivational. The theme of communication is frequently identified in research about barriers and facilitators to accessing prehabilitation [5–7], and poor information delivery can hinder patient involvement [8–10]. However, communication remains somewhat overlooked in the research literature about prehabilitation [11], and as such, understanding of how best to communicate with patients about prehabilitation remains incomplete.

Prehabilitation is typically delivered soon after patients are diagnosed with cancer. The psychological stresses of a cancer diagnosis are well-documented. Patients undergoing prehabilitation before cancer surgery have described feeling both emotionally overwhelmed and overloaded with information during this time window [8–10]. Some patients said they were unable to sufficiently retain or process the additional information given to them about prehabilitation at this time. This prevented them from engaging in future prehabilitation interventions [8–10].

It is common for patients to seek further information from publicly accessible sources as an alternative or supplement to information from healthcare professionals. Digital tools, websites, mobile applications and multimedia now represent a significant proportion of the information accessed by patients [12]. However, a survey commissioned by the Patient Information Forum found that half of UK adults struggled to access trusted health information with 1 in 10 stating that they had been affected by misinformation [12]. Misinformation about cancer treatment has the capacity to cause harm; for example, patients may decline evidence-based interventions, adversely impacting their physical and psychosocial health [13].

Providing credible information to patients about prehabilitation may be challenging. Evidence supporting the efficacy of prehabilitation interventions in improving certain outcomes of cancer treatment remains uncertain: different interventions appear to benefit patients with different cancer types differently [11,14,15], interventions are

delivered in different locations, formats and intensities across the prehabilitation research literature, numerous outcome measures have been used to evaluate the efficacy of these interventions [14,16], and the definition of prehabilitation even appears to be under consideration within the academic community [17]. Overall, this presents a challenging body of evidence to synthesise and summarise when aiming to communicate to patients in an honest and accessible way.

Our interest in the way that prehabilitation is communicated to patients was stimulated by working on the 'Prehabilitation Before Cancer Surgery: Quality and Inequality' (PARITY) study (NIHR 134282). During the PARITY study, we were exposed to public- and patient-facing literature and identified the tensions between evidence, accessibility and motivation; we noted a tendency for resources to promote the benefits of prehabilitation without recognition of the uncertainty in the clinical evidence base for interventions. Most strikingly, we found examples where prehabilitation was presented in the information as a means of reducing cancer recurrence [11]. We therefore designed a supplementary study which aimed to further investigate the role of online patient information resources in prehabilitation intervention delivery. We also considered whether the resources would reflect how prehabilitation services acknowledge the different backgrounds and circumstances of the patients they serve.

## Method

We used discourse analysis to investigate the language used in prehabilitation resources. Discourse analysis encompasses a range of different analytical approaches [18]. Common to these is understanding how language creates and mediates social and psychological realities. Discourse analysis is aligned with social constructionism and never treats language as neutral, transparent or solely for communication [19]. Documents may be viewed as the physical traces of the social practices of individuals and organisations [20]; analysis of the construction and function of these accounts therefore enables an exploration of the social practices with which they are associated. We wanted to understand how the language used by resources would navigate the tension between evidence, motivation and accessibility in the context of prehabilitation, including how the language used in resources promoted equitable access to services.

### Reflexivity

We approach this work from various perspectives: SS and CS are anaesthetists working in the NHS and this work means that they regularly encounter patients who are undergoing cancer treatment, though prehabilitation is not part of their regular clinical practice. CS is also a researcher with an interest in health inequalities, and has relevant lived experience as a carer for relatives undergoing cancer treatment. HS is a sociologist researching cancer care, inequalities and communication, with lived experience of cancer. We drew on our professional backgrounds and experiences in developing the study and analysing the data, but also attempted to avoid undue influence by maintaining an awareness of their potential impacts, and by discussing the study as a team as data were acquired and analysed.

### Ethical considerations

Formal ethical approval was not required for research involving publicly available information. However, this work was conducted in line with the Lancaster University research ethics policy and with reference to guidance from the Association of Internet Researchers [21]. It is impossible to fully anonymise quotes from freely available online material however, we have chosen to draw focus to the content rather than the institutions by using pseudonyms throughout. This work has been reported in accordance with the Standards for Reporting Qualitative Research [22].

### Data collection

A systematic grey literature searching strategy was adapted after consultation with a university librarian to locate relevant online patient information resources [23]. The three groups of resources searched were a database (TRIP

Pro, Trip Database Ltd, Newport, UK), search engines (Google, Google LLC, Mountainview, CA, USA and Bing, Microsoft Corp, Redmond, WA, USA) and websites identified by the research team as likely to contain relevant material. Specificity was improved by using search engines customised to website domains relevant to UK prehabilitation organisations and restricting results to the UK only. Discourse analysis is the study of language in its social and cultural context. We chose to conduct this study within a common language (e.g., UK English) and a 'national' service (e.g., the National Health Service) to ensure socio-cultural and linguistic consistency. The websites of cancer charities registered by the UK Government Charity Commission [24], trusted patient information websites listed by NHS England [25], and the websites of NHS trusts known to provide a prehabilitation service were searched [26]. Searches were conducted between 3/7/24 and 9/7/24. All search strategies involved three groups of terms: 1) prehabilitation, 2) cancer and 3) patient information. Searches were built with synonyms using non-technical language as the aim was to retrieve patient-facing resources. Detailed search strategies are provided in Appendix S1. [Subxref1]

Complete screening of the innumerable web pages returned by a search engine is an unmanageable task. Consistent with previous grey literature search reports, we agreed in advance to screen the first 100 results per search [23]. The position of a website within a search relies on the relevancy ranking algorithm within the search engine to highlight the most accessed results. This was beneficial for this project as the results most accessible to the researcher were also likely to be those most accessed by patients. A single investigator (SS) screened the search results, at the time of searching based on the website title and 'snippet' text beneath, against pre-specified inclusion criteria (Table 1). The full webpages were screened again for inclusion in the final analysis. The final list of included resources was agreed by all reviewers.

Individual webpages were grouped into cases where appropriate. A case represented a single organisation and all the relevant information produced by them and made available on their website. Cases were then categorised by organisation type that produced the material. Geographical and professional relationships between organisations were recorded.

**Quality appraisal**

The Quality Evaluation Scoring Tool (QUEST) was used to assess the quality of the included sources [27]. The tool assesses online health information across six domains (authorship, attribution, conflict of interest, currency, complementarity and tone) to determine the overall quality of a resource on a 28-point scale where a higher score indicates a higher-quality resource. It aims to appraise information quality from an expert perspective as opposed to assessing user experience or understandability for patients. Scores for the included resources were calculated by a single researcher (SS) and recorded in a spreadsheet (Excel, Microsoft Corp, Redmond, WA, USA).

**Sampling**

A purposive maximum variation sampling strategy was developed iteratively based on organisation characteristics and quality appraisal scores. The final sample satisfied the following criteria:

- Contained information from at least one of every organisation type and (where possible) in approximately the same proportions as the full data set.

- Contained information from every nation of the UK.

- Contained approximately even numbers of sources from each quartile according to the quality appraisal score.

- Organisations that were associated professionally or geographically were not represented more than once.

**Table 1. Inclusion and exclusion criteria.**

| Inclusion | Exclusion |
|---|---|
| Published by a UK organisation | Published by an organisation outside of the UK |
| Published by a reputable organisation, including but not exclusively:<br>- NHS<br>- UK registered charity<br>- Royal college/ recognised medical society/ organisation | Media articles, opinion pieces, blogs, newsletters, policy documents, social media, guidelines |
| Most up to date version of page | Historic or archived version of the webpage if duplicate page retrieved |
| Intended for patients or relatives, e.g., non-technical language used | Information for professionals, service commissioners, students, researchers, governments, e.g., contains technical or evaluative information |
| About prehabilitation or a prehabilitation service. Information may include but not exclusively:<br>- What is it?<br>- Who is it for?<br>- Benefits?<br>- Evidence<br>- Service information<br>- Education about specific aspects, e.g., exercise/ nutrition/ psychology/ stopping smoking | Information about:<br>- Preoperative assessment only<br>- General preoperative preparation, e.g., what to bring to hospital<br>- Generic healthy living advice/ getting ready for surgery not connected to a specific prehabilitation service prior to cancer treatment |
| Prehabilitation services aimed at cancer patients:<br>- Preparation prior to cancer surgery, radiotherapy or chemotherapy | Targeted prehabilitation for specific outcomes or not related to cancer treatment:<br>- Prehabilitation exercises prior to joint replacement<br>- Liver reducing diet prior to bariatric surgery<br>- Pelvic floor exercises prior to prostate surgery<br>- Surgical prehabilitation that does not specifically reference cancer |
| Webpages, videos,.pdf documents | |

Table 1 shows the pre-specified inclusion and exclusion criteria used to determine which resources should be considered for inclusion in the final analysis.

The concept of information power was used to estimate the approximate sample size that would be required for analysis at the time of study design [28]. Information power was considered throughout the study to determine when to conclude the analysis phase. Initially, sample size was considered in relation to a published study using similar methodology [29]. We determined our study to have a narrower aim and a more specific sample than this prior work, and therefore initially anticipated fewer resources would be required for our analysis. The initial search results confirmed the specificity of the sample, with resources typically being produced by similar organisation types (NHS Trusts or charities) who often shared the same content. The analysis further supported the use of a smaller sample as many resources contained similar themes which we were able to link to existing theory about behaviour change techniques and patient information discourses.

Individual webpages from sample cases that explained prehabilitation or a service were extracted into qualitative data analysis software (NVivo, Lumivero LLC, Burlington MA, USA) for analysis.

Exports were quality checked against the original site to ensure content was not distorted, altered or updated. The transcripts for multimedia content were extracted directly from webpages (if available) or created using dictation software (Word, Microsoft Corp, Redmond, WA, USA). All transcripts were checked for quality against the original content for analysis. Duplicate content (often videos) shared between organisations was only analysed once.

## Data analysis

Analysis followed the procedural stages outlined by Rapley [19]. Text was read sceptically from a perspective which foregrounds the constructive and functional properties of language, prompted by a series of questions outlined by Rapley and Willig [18,19]. A single researcher (SS) coded text inductively, working through the documents line by line. Codes were collated into a codebook which was refined iteratively using a constant comparison method. Data were analysed for similarities or variation in the language used to present concepts. Theories were formed and refined with reference to the data set, academic literature and discussion between the research team members.

## Results

Applying the search strategy yielded 3394 results from which 215 potentially relevant records were identified. After eligibility screening, records from 68 unique organisations remained (Fig 1). Records were produced by NHS trusts (n = 44), charities (n = 11), cancer alliances (n = 5), partner organisations (i.e., which deliver prehabilitation in the context of a formal affiliation to a Cancer Alliance or NHS trust) (n = 6), a patient information organisation (n = 1) and a professional organisation (n = 1). Information was retrieved directly or via related services (associated NHS trusts or partner organisations) from all but two of England's 21 Cancer Alliances; two of the three Scottish Cancer Networks; and the Welsh and Northern Irish Cancer Networks. Quality scores according to QUEST ranged from 7 to 28. The median score was 10 (IQR = 10–13). All sources scored fully in the *"complementarity"* and *"conflict of interest"* domains. Few sources scored well in the *"authorship"*, *"attribution"* and *"tone"* categories. Quality appraisal scores informed a purposive sample of 25 sources which met our predetermined criteria. A full breakdown of the quality appraisal scores is provided for the analysis sample (Fig 2) and non-sampled organisations in S2 Appendix.

Explanations of why and how patients should undertake prehabilitation were framed similarly by most organisations. Many used influential (as opposed to informative) phrases when explaining why patients should participate in prehabilitation. This focused on individual motivation, with less acknowledgement of personal resources, when explaining how patients could achieve the benefits of prehabilitation. These ideas overlap to suggest a belief that an individual's motivation is a key determinant of the success of prehabilitation and as such, organisations construct their information in a way that is designed to influence this.

## Analysis: Influence vs Inform

Almost all of the online patient information stated the benefits of prehabilitation. Many different benefits were described in individual resources and across the sample (Box 1). In contrast, the limitations or drawbacks of prehabilitation were rarely discussed. At most, a small number of organisations acknowledged contexts in which the benefits of prehabilitation were uncertain (Box 2). The benefits were mostly presented with certainty or emphasis (Box 3). Although some organisations adopted a more cautious tone with references to chance, risk or likelihood (Box 4), this nuance was lost when more certain language was used in other parts of the same sentence. For example, in the phrase "*you're less likely to suffer from complications*", "*you're*" [a contraction of 'you are'] indicates certainty but *"less likely"* acknowledges chance. The overall effect could reasonably be interpreted as a direct and certain relationship existing between intervention (prehabilitation) and outcome (complications). The benefits of prehabilitation were emphasised further by some organisations who included endorsements from patients, healthcare professionals and others in their online information (Box 5). Non-specific promotional statements were used by a few organisations (Box 6). However, many organisations used phrases such as "*evidence suggests*" or "*research says*" which adds legitimacy to the benefits stated (Box 7). These phrases provide authority whilst simultaneously reassuring the reader that the benefits of prehabilitation were tangible. The benefits are validated externally, rather than being solely based on the individual, albeit expert, opinion of the authors. The benefits were further evidenced by the inclusion of patients' and healthcare professionals' experiences (Box 8). These techniques

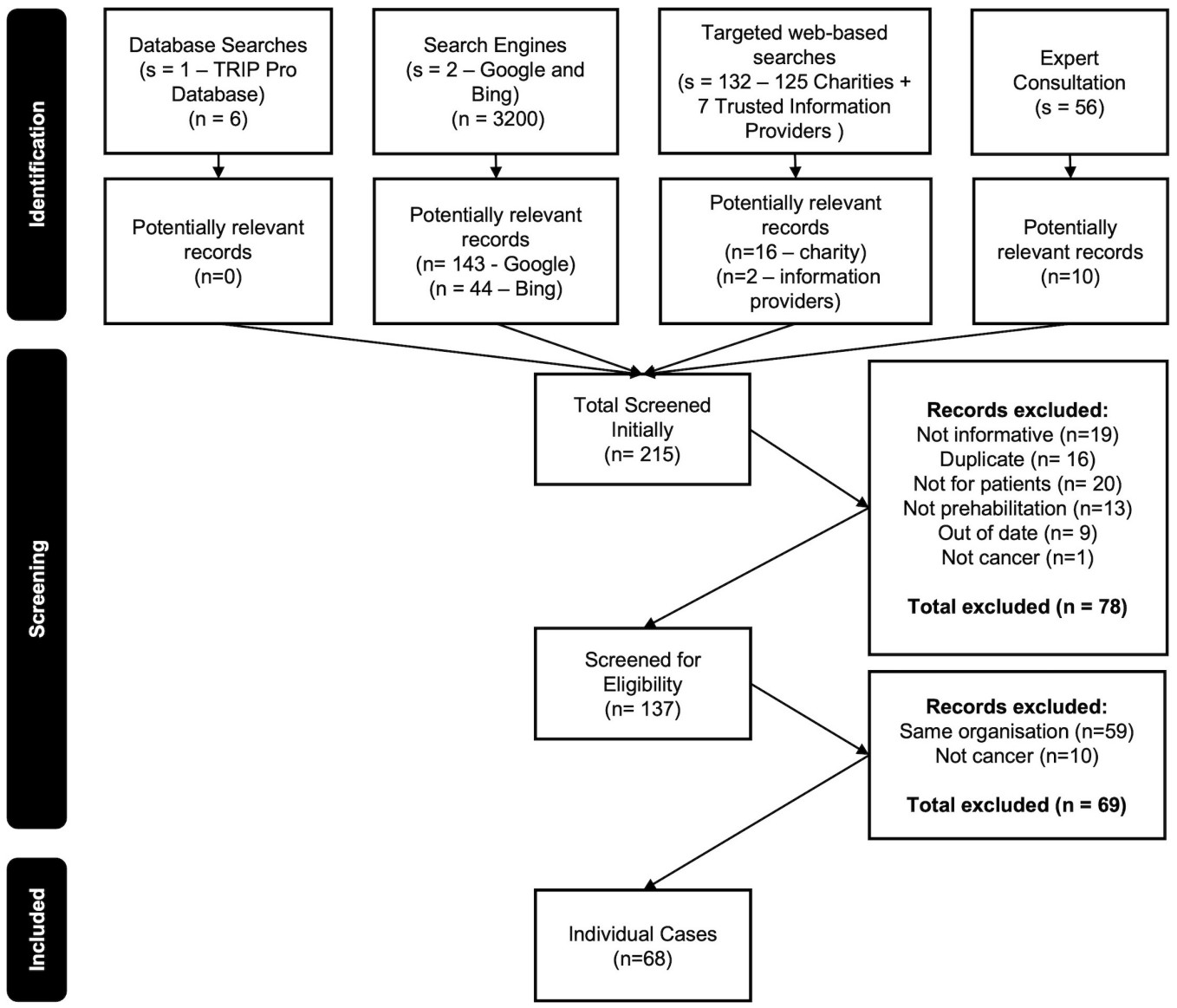

**Fig 1. Study flow diagram.** Search and screening phases of the study, including the number of records included in the final sample.

serve to strengthen the legitimacy of the information provided. The benefits are not only stated but can be demonstrated by research and/or experiences.

Two organisations presented the benefits of prehabilitation differently in their online information. NHS Trust 23 was unique in not listing any benefits, and NHS Trust 44 provided numerical data from their own service to illustrate its impacts. Whilst this organisation may have selected data to support their intervention, they also presented it alongside a limitation (*"these improvements are likely to be multifactorial"*). This suggests that the organisation may support the reader to form their own conclusions about whether the intervention was worthwhile or not. This contrasts with the other organisations in the sample which appear to assume that the benefits must be emphasised, evidenced and endorsed for patients to choose to participate.

**Fig 2. Quality appraisal scores.** Breakdown of the quality scores allocated using the Quality Evaluation Scoring Tool (QUEST) for the resources included in the analysis sample. [27].

| SOURCE | Authorship | Attribution* | Attribution 2 | Conflict of Interest | Currency | Complementarity | Tone* | Total + |
|---|---|---|---|---|---|---|---|---|
| NHS Trust 2 | 0 | 0 | 0 | 2 | 0 | 2 | 0 | 7 |
| NHS Trust 4 | 0 | 0 | 0 | 2 | 0 | 2 | 0 | 7 |
| Partner 1 | 0 | 1 | 0 | 2 | 0 | 2 | 0 | 10 |
| NHS Trust 8 | 0 | 1 | 0 | 2 | 0 | 2 | 0 | 10 |
| NHS Trust 14 | 0 | 1 | 0 | 2 | 0 | 2 | 0 | 10 |
| NHS Trust 15 | 0 | 1 | 0 | 2 | 0 | 2 | 0 | 10 |
| Partner 4 | 0 | 1 | 0 | 2 | 0 | 2 | 0 | 10 |
| Cancer Alliance 2 | 0 | 1 | 0 | 2 | 0 | 2 | 0 | 10 |
| NHS Trust 23 | 0 | 0 | 0 | 2 | 0 | 2 | 1 | 10 |
| NHS Trust 24 | 0 | 1 | 0 | 2 | 0 | 2 | 0 | 10 |
| Charity 3 | 0 | 0 | 0 | 2 | 2 | 2 | 1 | 10 |
| Charity 5 | 0 | 1 | 0 | 2 | 2 | 2 | 0 | 12 |
| NHS Trust 27 | 0 | 1 | 0 | 2 | 2 | 2 | 0 | 12 |
| Cancer Alliance 5 | 0 | 1 | 0 | 2 | 2 | 2 | 0 | 12 |
| NHS Trust 29 | 0 | 1 | 0 | 2 | 2 | 2 | 0 | 12 |
| NHS Trust 30 | 0 | 1 | 0 | 2 | 2 | 2 | 0 | 12 |
| NHS Trust 31 | 0 | 1 | 0 | 2 | 2 | 2 | 0 | 12 |
| Professional 1 | 1 | 1 | 0 | 2 | 2 | 2 | 0 | 13 |
| NHS Trust 35 | 1 | 1 | 0 | 2 | 2 | 2 | 0 | 13 |
| NHS Trust 36 | 0 | 1 | 0 | 2 | 0 | 2 | 1 | 13 |
| NHS Trust 40 | 1 | 1 | 0 | 2 | 2 | 2 | 0 | 13 |
| NHS Trust 43 | 2 | 2 | 0 | 2 | 2 | 2 | 0 | 17 |
| NHS Trust 44 | 0 | 1 | 0 | 2 | 2 | 2 | 2 | 18 |
| Charity 11 | 0 | 3 | 2 | 2 | 2 | 2 | 2 | 26 |
| Information 1 | 2 | 3 | 2 | 2 | 2 | 2 | 2 | 28 |

**Legend**

- Scores 3
- Scores 2
- Scores 1
- Scores 0

*Attribution and Tone scores multiplied by 3

+Total score is out of a possible 28

**Box 1**

*"Scientific information tells us prehabilitation can help you to:*

*Reduce your anxiety and improve your mood.*

*Improve your energy levels, reducing tiredness.*

*Maintain your independence and do more of your normal day to day activities.*

*Improve your sleep pattern.*

*Improve your general fitness and sense of wellbeing.*

*Have fewer problems during your cancer treatment.*

*Have fewer admissions to hospital and shorter stays in hospital.*

*Have a better response to your future cancer treatment.*

*Recover more quickly from treatments.*

*Lower your chances of cancer coming back after any treatment finishes.*

*Help you manage any other medical or health conditions.*

*Promote long-term healthy lifestyle and well-being."*

- NHS Trust 40

**Box 2**

*"Research shows that it is helpful for people having surgery. But researchers think it might also help people having other cancer treatments such as chemotherapy and radiotherapy."*

- Charity 11

*"…authors found that although there was a strong evidence basis for non-cancer prehabilitation, we're not yet there with the cancer-specific evidence."*

- Information Organisation 1 communicates the findings of a prehabilitation systematic review to readers

**Box 3** (formulations identified by the researchers that imply certainty or emphasis are denoted in bold type in the quotations below)

*"We **know** that preparing for your treatment **will** help you to recover quicker, alongside supporting your longer-term health and wellbeing too."*

- NHS Trust 4

> "we **know** that **actually** now **there is** a reduction in the chance of cancer recurrence if you maintain your fitness throughout your cancer journey."
>
> - Healthcare professional speaking in Partner 1 video
>
> "Improving your health and fitness before surgery **has been shown** to have **huge** benefits, particularly for people who may be about to have surgery for cancer."
>
> - NHS Trust 2

## Box 4

> "you're less likely to suffer from complications"
>
> - NHS Trust 2
>
> "…reduce your chance of developing a chest infection"
>
> - NHS Trust 8

## Box 5

> "If I was talking to somebody who was recommended to have prehab, and they were unsure about it, I would say definitely, definitely go for it. Definitely."
>
> - A patient speaking in a Partner 4 video
>
> "Dr [name anonymised] is a GP and also the clinical lead for cancer for [location anonymised] CCGs, and says that anything that helps patients become fitter and more healthy in anticipation of possible treatment for cancer would be something that he would support."
>
> - Information 1
>
> "We are national award-winning NHS service"
>
> - Partner 1

## Box 6

> "So you can see why exercise is sometimes referred to as underrated wonder drug."
>
> - A healthcare professional speaking in an NHS Trust 15 video
>
> "If exercise was a pill it would be prescribed to every cancer patient"
>
> - Partner 1

**Box 7**

*"There is unmistakable evidence that prehabilitation helps to improve stamina prior to having treatment and aids recovery."*

- Cancer Alliance 2

**Box 8**

*"Many thanks for the brilliant exercise classes, the care, the fun and the laughter that has kept me going over this difficult period and got me to a good level of fitness after my first operation and to cope with the next stage of my recovery. It is truly wonderful work you all do."*

- Written patient quotation provided by Partner 1

*"Afterwards, you can definitely tell the patients that are going through the program. They're keen to get up and going, and they asked me when they're going home, when they can go back to the gym, which they can't do just at the time but they're but they're keen and that and I think that's a really good sign."*

- A healthcare professional speaking in a Partner 4 video

Organisations recognised the anxiety and fear associated with cancer diagnosis and treatment in their online information (Box 9). Patients may be motivated to participate in prehabilitation because they are told that it can help them to manage some of the negative emotions they may be experiencing. For example, some organisations described how at the time of cancer diagnosis, patients may feel out of control (Box 10). A common benefit listed by several organisations was the sense of control that prehabilitation provides (Box 10). Online information also stated that prehabilitation could directly improve some mood symptoms (Box 11). Prehabilitation services commonly offered to provide support. The *"support"* offered could broadly be divided into emotional or interventional support. Emotional support often addressed themes of isolation by offering friendship or contact with others (including professionals) (Box 12). Prehabilitation is again positioned as an intervention to manage distressing feelings (such as loneliness) which may provide motivation for patients to access a service.

**Box 9**

*"You may be feeling overwhelmed or even scared by the thought of going on treatment."*

- Charity 3

**Box 10**

*"So it's understanding that people often feel very out of control when they're first diagnosed with cancer and in the run up to treatment."*

- A healthcare professional speaking in a Partner 4 video

*"It can also help you to feel more in control of what's happening."*

- NHS Trust 30

> **Box 11**
>
> *"Getting ready for treatment will help you feel prepared and stop the feelings of anxiety or depression from getting worse."*
>
> - A healthcare professional speaking in Charity 11 video

> **Box 12**
>
> *"We understand that making these changes might feel like a lot, especially when you're dealing with the impact of cancer. But you don't have to do it alone. If you need help or support, we're here for you."*
>
> - NHS Trust 30
>
> *"It also provided a community of lovely people who provided support and friendship. I can honestly say I don't think I would be in as strong a position today without it."*
>
> - Written patient quotation provided by Partner 1
>
> *"Group sessions can provide a support network of people who understand what you are going through."*
>
> - NHS Trust 35

The negative emotions associated with cancer diagnosis may have the potential to motivate engagement in prehabilitation in another way. For example, fear of morbidity or mortality may motivate patients to adhere to an intervention which promises to reduce the likelihood of these consequences. Some information provided to patients appeared to capitalise on and amplify this fear to encourage the uptake of prehabilitation interventions.

Organisations suggested that prehabilitation may improve recovery or reduce complications; side-effects; or length-of-stay in hospital. These aspects of cancer treatment may be particularly worrying for patients [30]. Moreover, the information often referred to the ability of prehabilitation to help patients cope with, withstand or tolerate cancer treatments. Patients were reminded throughout the information that cancer treatments were challenging, stressful and daunting (Box 13), which has the potential to amplify patients' fear about impending interventions. Authors could have chosen to present cancer treatment in non-emotive or even reassuring ways, and their decision to do otherwise suggests that amplified fear may serve a purpose.

Several organisations compared prehabilitation to training for a marathon (Box 14). Whilst this commonly-used analogy is perhaps intended to illustrate the need for preparation, it also positions cancer treatment as being exceptionally physically demanding, and in the context of concerns about their physical health or fitness, patients may find the idea of running a marathon to be particularly daunting. Furthermore, organisations acknowledge the short time-frame that patients may have between diagnosis and treatment. Considering that preparing for a marathon typically takes many months, the idea of having to prepare for this in a few weeks may exacerbate the sense of stress and urgency. Overall, we question the usefulness of the 'marathon' analogy. Unlike running an actual marathon, patients will not have made a positive choice to undergo cancer treatment, are unlikely to feel that they have the option to decide not to proceed or choose an easier alternative, and are unlikely to have the opportunity to delay interventions to enable them to prepare fully.

**Box 13**   **(formulations identified by the researchers that emphasise the difficulty of cancer treatment denoted in bold type in the quotations below)**

*"It [prehabilitation] can be particularly helpful to patients with cancer as increased fitness helps them to **cope with stress of treatments** like surgery, chemotherapy, or radiotherapy."*

- NHS Trust 36

*"This is the time you can make the difference before **surrendering** to the medicine..."*

- Written patient quotation provided by Charity 5

*"Chemotherapy and radiotherapy can have a **devastating** impact on the body especially for patients who may have ill health to begin with"*

- Written healthcare professional quote provided by Information 1

**Box 14**

*"Prehab helps you to be fitter and stronger, like training for a marathon."*

- NHS Trust 35

*"The stress of a major surgery is like running a marathon - both require preparation and training"*

- NHS Trust 43

*"Going through cancer treatment can feel like a marathon, and you wouldn't run a marathon without training beforehand."*

- Partner 4

Organisations suggest prehabilitation can improve the outcomes, results or success of treatment, with some saying it can reduce the risk of cancer recurrence or future cancers developing (Box 15). Some organisations also state that prehabilitation may increase options for treatment. Other phrases such as "*do better*", "*improved outcomes*" or "*treatment results*" are vaguer. Yet when used in contexts where health literacy is unequal, these phrases may similarly be interpreted to mean successful cancer treatment (cure or remission) as opposed other consequences such as fewer complications. Similarly, phrases which allude to the future imply that prehabilitation will also help patients to live well *beyond* cancer or improve their *long-term* health (Box 16). Whilst there may be an association between lower cancer incidence in populations with higher levels of physical activity, to suggest that prehabilitation reduces cancer recurrence is an overstatement of the current evidence base [31]. Arguably, treatment failure and survival are some of the most emotionally challenging concepts that patients may be confronted with, and therefore an individual's motivation to engage with any interventions that promise to influence these outcomes may be strengthened by possibility of reducing this risk.

> **Box 15**
>
> *"Prehab aims to improve your treatment results and reduce any side effects which may slow down your recovery."*
>
> - NHS Trust 35
>
> *"The benefits of prehab… reduces the chances of your cancer returning, or the chances of other cancers and health conditions developing"*
>
> - Cancer Alliance 5
>
> *"…we understand that having a cancer diagnosis is really devastating for patients but we know that if we can get you as fit as possible before your treatments that you'll do better…"*
>
> - Healthcare professional speaking in Partner 1 video

> **Box 16**
>
> *"Prehabilitation helps to build resilience to treatment by reducing some of the common side effects, and improving your long-term health"*
>
> - NHS Trust 29

The online information promoted the benefits of prehabilitation whilst rarely discussing the limitations or alternatives. Emotion was used to strengthen the value of the benefits described. Overall, the resources worked to convince patients of the advantages of prehabilitation as opposed to empowering them to make an informed choice about why they should participate.

### Analysis: Motivation vs Resources

The information provided by prehabilitation organisations more often focused on influencing individual motivation as opposed to recognising wider determinants that may affect participation in a service. The information focused on addressing perceived psychological barriers whereas practical barriers were rarely discussed. Organisations referred to the personalisation of interventions. However, this mostly referred to matching interventions to the physical needs of a patient as opposed to their social circumstances. Lastly, the information worked to emphasise the individuals' responsibility for performing the interventions. These themes align to contribute to the idea that individual motivation is the main determinant of participation in prehabilitation and as such the information focuses on influencing this.

Potential concerns that patients could have about participating in prehabilitation were addressed in the online information. Predominantly it focused on psychological as opposed to practical barriers to participation. The online information provided reassurance to patients about the knowledge, fitness levels or time investment required to experience the benefits of prehabilitation (Box 17). Academic literature recognises the short timeframe between cancer diagnosis and treatment, during which there are multiple competing demands on a patient's time, as a specific barrier to participation in prehabilitation [6,7]. Some organisations may have sought to include and encourage patients to participate during this window by addressing these concerns in their online information.

However, the way in which some organisations addressed these concerns suggests that they perceive these barriers to be more related to individual motivation than an understanding about their interventions. Some organisations go beyond

providing information to reassure and instead attempt to use language to motivate. The efficacy of prehabilitation for patients who had limited time to complete interventions was sometimes addressed using the phrase *"as little as"* (Box 18). the function of this phrase is to emphasise – to go beyond informing patients that prehabilitation in a short timeframe was beneficial.

In another phrase that occurred frequently "*small changes*" was positioned next to "*big difference*" (Box 19). this 'value proposition' acts to minimise the potential commitment whilst maximising the potential benefits of prehabilitation. The simplicity of prehabilitation was also emphasised in phrases such as *"every little helps"* (Box 20). these phrases all appear to go beyond informing patients about the time or actions required to experience the benefits of prehabilitation. By emphasising the simplicity of prehabilitation, they provide greater assurance that it is within the readers ability to attain the benefits potentially motivating them to adhere to the intervention.

### Box 17

*"If you are not currently very active or feel you could do more, this is the perfect time to start."*

- NHS Trust 29

*"This includes an introduction to the gym by a trainer and specific exercise instructions to support you."*

- NHS Trust 23

*"Sometimes, you might not have much notice before your treatment starts, or you might find making these changes overwhelming. Just do what you can and be kind to yourself."*

- Charity 3

### Box 18

*"The benefits of prehabilitation can be seen in as little as two weeks."*

- NHS Trust 14

### Box 19

*"Making small changes to your lifestyle, can make a big difference to the way you respond and recover from cancer treatment."*

- Cancer Alliance 5

### Box 20

*"Even small changes can help, so start slowly and aim to gradually build up activity."*

- NHS Trust 27

*"Every little helps!"*

- Charity 5

Practical barriers to participation in prehabilitation were less frequently addressed in the online patient information. The safety of exercising with cancer was sometimes assured (Box 21). The academic literature has previously identified limited finances or difficulty travelling as barriers to accessing prehabilitation [6,7]. Some services stated that their interventions were free or provided in a choice of formats or locations (Box 22). these examples demonstrate a potential awareness within some organisations of the non-motivational barriers that may prevent patient participation in prehabilitation.

> **Box 21**
>
> *"Research evidence suggest that it is safe to exercise in the weeks before surgery and during radiotherapy and chemotherapy."*
>
> <div align="right">- NHS Trust 43</div>

> **Box 22**
>
> *"[Prehabilitation service] is provided in leisure facilities across [location], which means that often patients can access the service close to their home."*
>
> <div align="right">- Partner 4</div>
>
> *"There is specialist support available to you free of charge to help you improve your fitness."*
>
> <div align="right">- NHS Trust 31</div>
>
> *"If you can't attend a workshop, you can get the same advice form a support and information specialist in person or over the phone."*
>
> <div align="right">- NHS Trust 30</div>

The individualisation of prehabilitation interventions was a frequently occurring theme in the patient-facing literature (Box 23). When organisations referred to the personalisation of interventions, this commonly meant adjustments to suit the biomedical needs of a patient (Box 23]. Fewer organisations referenced the personalisation of interventions to suit patients' socio-economic circumstances, cultural preferences or personal choice (see Box 24 for one notable exception). The focus on tailoring interventions to a patient's physical attributes is another way in which the information conveys the feasibility of prehabilitation. The information works to reassure patients that they will not be expected to do interventions that are beyond their physical capacity, aiming to encourage them to participate. Again, fewer organisations seem to acknowledge the need to tailor interventions to patient's circumstances or choices.

> **Box 23**
>
> *"If you are referred to [prehabilitation service], the team will work with you to produce a tailored programme to your specific health needs."*
>
> <div align="right">- NHS Trust 36</div>
>
> *"A personalised exercise programme will then be agreed based on the results of the tests."*
>
> <div align="right">- NHS Trust 44</div>

> "Our clinical exercise physiologists will assess your fitness and work with you on a personalised plan to improve your fitness before surgery."
>
> - NHS Trust 23

**Box 24**

> "If you are new to exercise, then have a think about what could work for you and would fit within your routine. You are much more likely to stick to an activity plan, if it is something you enjoy doing or fits into your day."
>
> - NHS Trust 24

The information emphasised the importance of the individual in prehabilitation interventions. Some organisations linked individual participation and intervention success explicitly (Box 25). More implicitly, organisations chose to address individuals directly in their information. Sentences emphasised the position of the individual as central to the intervention and as the recipient of the benefits (Box 26). Phrases such as *"self-care"*, *"self-manage"* or *"take care of yourself"* similarly emphasise organisations' expectation that it is the individual's responsibility to act. This was also implied by a phrase common to some organisations that suggested individuals *"take an active role"* in their care. A few organisations included comparisons to others in their information (Box 27). These statements set out the expectation that services have of patients. That expectation is one of participation or compliance.

As stated above, support was offered by many online patient information resources (Box 28). This appears to contradict the idea that prehabilitation is solely the responsibility of the individual, particularly when emotional support often addressed feelings of isolation. Support was also offered in the context of interventions although this was often phrased in a way that maintained the assumption of individual responsibility (Box 28). Support comprised information or guidance, but actions remained the responsibility of the individual.

**Box 25**

> "YOU [sic] are most important- what you do now can have a big impact on your recovery"
>
> - NHS Trust 8

> "This is a long, positive list of benefits, but they can only happen if you take part in prehab in a meaningful way."
>
> - Charity 5

**Box 26**

> "This leaflet provides information and support on how you can prepare for your treatment, helping you to take control and be involved in your care."
>
> - NHS Trust 40

**Box 27**

> "Many people are glad to know they can do something immediately to help their health."
>
> - NHS Trust 2 and NHS Trust 14

> **Box 28**
>
> "In this video, we would like to support you by showing what you can do to prepare yourself and explaining why preparation is so important."
>
> <div align="right">- Healthcare professional speaking in Charity 11 video</div>
>
> Your pre-operative physiotherapy team can support you with making your plan if required.
>
> <div align="right">- NHS Trust 43</div>

## Discussion

Prehabilitation is positioned within patient-facing information as a simple and feasible intervention that confers significant and certain benefits; the information is written in a way that works to convince the reader of this. It frames a patient's decision to participate in prehabilitation as a cost-benefit calculation. The information works to influence this calculation in favour of participation. The information less frequently acknowledges the limitations or alternatives to prehabilitation. Emotive framing of the consequences of non-participation further strengthens the argument for doing prehabilitation. It is implied that an individual patient can determine the success of prehabilitation interventions if they engage sufficiently. The information seeks to activate patients, and focuses on individual responsibility and less frequently acknowledges systemic and structural barriers that prevent involvement. Patients' motivation is discursively positioned in the information as the main determinant of engagement in prehabilitation.

Promoting prehabilitation in public-facing literature is appropriate if it increases participation in a beneficial intervention. However, this is contingent upon two assumptions which are worthy of further consideration. Firstly, research needs to establish whether this communication strategy is effective at promoting participation in practice. Secondly, given the uncertainty surrounding the efficacy of prehabilitation in impacting cancer treatment outcomes, what constitutes a '*beneficial*' intervention for patients requires clarification. Whilst this analysis can explore these considerations, to improve future information resources, further research involving patients is required.

Current UK prehabilitation guidance references the Capability, Opportunity and Motivation model of Behaviour Change (COM-B) [3,4]. In comparison with earlier behaviour change models, this acknowledges the wider determinants of health related behaviour (such as *"social opportunity"* or *"physical capability"*) beyond *"individual motivation"* [4]. However, the way in which patient information resources address the *"individual motivation"* or *"psychological capability"* components of the COM-B model is informative. How these behavioural determinants are addressed may be more aligned with an earlier behaviour model with greater emphasis on individual motivation. The themes used to promote engagement in patient information are consistent the *"perceived benefits"* and *"perceived barriers"* dimensions of the Health Belief Model [32]. The *"teachable moment"* is a concept which developed from the Health Belief Model [33], and also appears in UK prehabilitation policy [3]. It describes a time when an individual may be more receptive to conversations about behaviour change (such as smoking cessation) because of a naturally occurring health event (such as cancer diagnosis or surgery) [3,33]. The Health Belief Model primarily focuses on individual perceptions of health and illness. The implication that patients make health-related decisions using a cost-benefit approach; the focus on individual motivation; and limited references to the wider determinants of health are all features identified in the prehabilitation patient information that suggest consistency with the Health Belief Model. By assuming that individual perceptions govern health behaviour, it is rational to promote the benefits and place the responsibility for change on the individual.

Designing patient information with reference to literature about behaviour change theory or the facilitators of service engagement does not actually establish whether these resources promote participation in practice. How patients

experience and respond to resources is not currently understood. Researchers have not established which communication strategies are most effective at motivating participation in prehabilitation. Behavioural scientists acknowledge the complexity of prehabilitation programmes, which aim to simultaneously change multiple aspects of behaviour (e.g., exercise, diet, smoking etc.) [2]. While the *'teachable moment'* has often been applied to change single *'unhealthy'* behaviours [33], its effectiveness in the context of a programme of complex behaviour change, such as prehabilitation, is unclear. The efficacy of motivating behaviour change using emotional affect, as the '*teachable moment'* does, is also questioned. Responses to a perceived health threat can include paralysing anxiety, denial or rebellion, which all may prevent an individual from engaging constructively with prehabilitation [34].

There are suggestions within the wider prehabilitation literature about the communication preferences of some patients which may not align with the approach taken by these resources. Most notably, Powell et al. observed that some patients valued a supportive, rather than directive, approach to communication [8]. They noted that when staff minimised pressure about participating in the prehabilitation programme, this paradoxically led to patients feeling more reassured and willing to take part. Powell et al. comment on the importance of healthcare professionals ensuring that when prehabilitation is communicated, patients feel they have a choice whether to take part [8]. A benefit of prehabilitation that is promoted and experienced by patients is the *"sense of control"* it provides during cancer treatment [6–8]. Providing patients with the choice of participating in prehabilitation may enhance the feeling of control that is perceived as beneficial. The findings from Powell et al. are of further importance as the study purposefully aimed to capture the experiences of prehabilitation non-engagers [8]. Current communication approaches in prehabilitation may not be motivating for all patients. Supportive communication techniques could encourage involvement from more individuals.

Studies of the barriers and facilitators to prehabilitation participation also identified factors relating to the other components of the COM-B model, which acknowledges determinants of behaviour beyond those related to individual motivation [6,7]. For example, barriers to engagement included physical symptoms (physical capability); insufficient time and finances (physical opportunity); and anxiety or stress (automatic motivation) [6,7]. Patient information resources are practically aligned with influencing certain determinants of behaviour more than others. Resources represent a direct communication between the patient and healthcare provider meaning that they are better placed to address psychological motivators through the provision of information. They are less able to influence barriers which affect physical opportunity. However, by focusing on individual motivation within information resources without recognising the wider determinants of health-related behaviour, prehabilitation services risk perpetuating intervention-generated inequalities [2]. Patient information resources that do not acknowledge the socio-economic challenges that some patients face, risk losing engagement from a group of individuals who feel they cannot match the expectations that providers have of them.

The second consideration raised by this analysis concerns how *'beneficial'* is being defined in the context of prehabilitation. Evidence that prehabilitation improves certain cancer treatment outcomes is mixed [11,14,15]. Some forms of prehabilitation may be more appropriately described as *potentially* beneficial to *some* patients in *some* situations. Our analysis suggests that instead of communicating this message, information resources promote the benefits of prehabilitation, without acknowledging it's limitations. To understand whether this approach is a problem, the potential costs associated with prehabilitation and harms of this communication strategy should be considered.

There is a potential opportunity cost associated with participating in prehabilitation. Research with patients about prehabilitation services has found that they often experience multiple competing demands in the window between cancer diagnosis and treatment [6,7,9]. Demands include attending multiple appointments; financial or employment preparations before treatment; and processing the burden of information and emotions related to a cancer diagnosis [6,7,9]. Prehabilitation takes place within this crowded time window and requires significant patient engagement. This places an additional burden on patients who must choose how to allocate their already strained time and resources. If aware of prehabilitation's limitations, patients may choose to prioritise other (meaningful) activities. To empower patients to make an informed choice about participation in prehabilitation, balanced information is needed.

Communication has been recognised as an important feature of cancer care (beyond prehabilitation) for its ability to impact treatment outcomes and experiences [35]. Shared decision-making not only promotes compliance, but also improves patients' experiences of treatment [35]. Information provision is recognised as a necessary for shared decision-making [35,36]. Yet establishing what information should be provided to support this process is more complex [35]. Patients' information requirements are unique to their individual circumstances, preferences and clinical situation [35,36]. Public-facing resources, such as those analysed in this study, are poorly positioned to address the information requirements of individual patients. Authors must therefore decide what information should be included that will be relevant to most audiences. In the context of prehabilitation, understanding the information that is relevant to most patients presents a challenge. There is limited research characterising which outcomes of prehabilitation are valued and prioritised by patients [11]. Research commonly evaluates prehabilitation efficacy using biomedical or clinician-oriented outcomes [11,15,16]. Previous research had shown that even when prehabilitation demonstrates improvement in clinician-oriented outcomes, this does not cause a corresponding increase in patient-reported quality-of-life scores [11,15]. Resources are therefore produced without knowing what patients want from prehabilitation services or how clinical benefits manifest as tangible improvements in treatment experiences.

Content considered relevant by authors may not align with information that is important to patients. This introduces a risk that resources simultaneously contain unnecessary information whilst omitting content relevant from the patient perspective. Authors may write information in a way that aims to encourage and enthuse patients about an intervention they genuinely believe will have significant benefits. Whilst this may have been done with good intentions, it risks providing information in a way that promotes compliance with clinician-oriented as opposed to patient-prioritised objectives. The idea that patient information leaflets less often empower patients to make an informed choice, instead opting to educate their decision-making so that it is in line with medical establishment objectives, has been recognised in academic literature previously [37]. Dixon-Woods characterises two discourses prevalent in the literature about patient information leaflets and explores the potential problems with and motivations for both [37]. Dixon-Woods argues that an *"education"* discourse presents non-compliance as the result of patients' incompetence [37]. It assumes that patients behave in a certain way only because they lack the knowledge to do otherwise. Leaflets consistent with the *"education"* discourse adopt the paternalistic approach of only providing certain *"necessary"* information. In contrast, the *"empowerment"* discourse assumes patients to be capable of weighing up information on healthcare effectiveness and to make decisions in line with their own needs and preferences. Our analysis of patient information identified how resources promote the benefits of prehabilitation without acknowledging it's limitations. This appears to be more aligned with an *"education"* as opposed to an *"empowerment"* discourse. For patients to be able to experience shared decision-making and its associated benefits in relation to prehabilitation, a communication approach aligned with the *"empowerment"* discourse may be a more appropriate.

Alongside identifying the opportunity costs of interventions, the potential harms of promoting prehabilitation and focusing on individual responsibility also require consideration. Our analysis identified how resources may give patients an expectation of avoiding the complications of cancer treatment through engagement in prehabilitation. At a population level, this may be true for some patients in some circumstances. However, for individual patients this is not certain, despite how the benefits of prehabilitation are presented in the resources studied. The usefulness of population-based statistical information for informing individual patient outcomes has previously been explored [34]. The challenge healthcare professionals face in communicating this information to individuals in a meaningful way is recognised [34]. Overpromoting the benefits of prehabilitation to individual patients living with cancer may carry psychological risk. It may instil unrealistic expectations about what prehabilitation can achieve which may risk introducing *"false hope"*. In the context of cancer, the implications of *"false hope"* have previously been explored when promoting lifestyle modifications using the *"teachable moment"* [34]. *"False hope"* may lead patients to prioritise prehabilitation over other aspects of their lives or treatments. It may skew patients' appreciation of the severity of their condition, or treatment risks, potentially influencing decisions they make about other aspects of their care. Psychological risk may also be introduced by leveraging prehabilitation as a route

to successful cancer treatment outcomes and by allocating responsibility to the individual for its completion. This could inadvertently introduce patient blame or guilt if a patient suffers a treatment complication. A patients' ability to cope with complications could be impacted if they feel somehow responsible.

## Strengths and limitations

To our knowledge, this is the first study to systematically search or analyse the patient-facing literature about prehabilitation. As such, a strength of this study is that it characterises prehabilitation services from a different perspective to previous work. This study uses naturalistic observation of online information to understand prehabilitation services. Naturalistic observation seeks to avoid the presence of the researcher influencing how organisations or participants behave and communicate, which may subsequently be reflected in the data collected. Organisations or individuals may choose to present themselves in a certain way that is different to their usual behaviour if they know that they are being studied.

A limitation of this study is the focus on online information. Whilst many patients now use the internet to obtain healthcare information; digital literacy and access to technology are barriers known to exist in the prehabilitation context [6,7,15]. As such, there may be many patients that this information does not reach meaning that the transferability of these findings to those contexts may be limited. In addition, organisations may also provide verbal or paper-based information to their patients, which this study did not analyse. Whilst there are advantages to using naturalistic observation to understand prehabilitation services, there are also limitations due to the inability to interact with the authors of the information. The meaning and intent of content in the online information must be assumed. There is no opportunity to interrogate the authors to understand why features of the information have been included, omitted or phrased in a certain way. Additionally, we cannot interrogate the intended readers to understand how they perceive the information and what consequences this may have. Without interaction with the authors or readers, our analysis relies on our own interpretations of the resources. This will be influenced by our individual experiences and biases. To address this, we discussed our findings as a multidisciplinary group and reflected on how our positions may have influenced our conclusions.

## Conclusion

The contextualisation of this discourse analysis raises several considerations about the framing of prehabilitation within the patient-facing resources. An approach which seeks to motivate compliance as opposed to empower a choice about participation in interventions may be problematic given current limitations in the available research. This analysis supports calls from the wider literature for a greater integration of behaviour science into the delivery of prehabilitation, extending to the patient information resources. Future research is needed to understand how patients' experience these resources and how this impacts their motivation to engage with interventions. Further research with patients is required to establish what they actually want from prehabilitation services. The co-creation of equitable, meaningful and high-quality resources between patients and services may be an important way of supporting the future delivery of prehabilitation before cancer treatments.

## Supporting information

**S1 Appendix. Detailed search strategy. This appendix provides the detailed search strategy and terms used.**
(DOCX)

**S2 Appendix. Quality appraisal scores. This appendix shows a breakdown of the quality scores allocated using the Quality Evaluation Scoring Tool (QUEST) for all the resources, including those that were not included in the final analysis. [27]. Attribution and Tone scores are multiplied by three. Total score is out of 28.**
(DOCX)

## Acknowledgments

The authors thank Lancaster University librarian John Barbrook for his advice on searching for grey literature. No other competing interests declared.

## Author contributions

**Conceptualization:** Sophie Stanley, Hilary Stewart, Cliff Shelton.

**Data curation:** Sophie Stanley.

**Formal analysis:** Sophie Stanley, Hilary Stewart, Cliff Shelton.

**Investigation:** Sophie Stanley.

**Methodology:** Sophie Stanley, Hilary Stewart, Cliff Shelton.

**Supervision:** Hilary Stewart, Cliff Shelton.

**Writing – original draft:** Sophie Stanley.

**Writing – review & editing:** Hilary Stewart, Cliff Shelton.

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
