## [Decision Letter · Decision Letter 0]

24 Sep 2025

PONE-D-25-37313Promoting or pressurising participation? A discourse analysis of online patient information resources about prehabilitation before cancer treatmentPLOS ONE?

Dear Dr. Stanley,

Thank you for submitting your manuscript to PLOS ONE. After careful consideration, we feel that it has merit but does not fully meet PLOS ONE’s publication criteria as it currently stands. Therefore, we invite you to submit a revised version of the manuscript that addresses the points raised during the review process.

1. Abstract: “Databases, search engines and websites (identified by experts)”…. Who is an expert?

2. This appears to be a systematic review, but the authors have not provided any evidence of prospective registration of the protocol. Hence, it is not possible to decide if the students followed the predetermined protocol. This makes it PRISMA non-compliant. Please clarify.

3. The Introduction runs into 4 pages and should be made more succinct. On the whole, the paper is very long and could be abridged.

4. Why did the authors limit their search to UK data? This limits generalisability.

5. There is no data-sharing statement.

6. All included studies/resources have not been referred to in the bibliography

We look forward to receiving your revised manuscript.

Kind regards,

Prateek Srivastav

Academic Editor

PLOS ONE

Journal Requirements:

“SS is an Academic Clinical Fellow whose salary is part funded by the National Institute for Health and Care Research (NIHR). The salary for HS was funded by the NIHR Health Services and Delivery Research programme as part of the PARITY study (NIHR134282). The views expressed are those of the authors and not necessarily those of the NIHR or the Department of Health and Social Care. The funder did not play any role in the study design, data collection and analysis, decision to publish, or preparation of the manuscript.”

“SS is an Academic Clinical Fellow whose salary is part funded by the National Institute for Health and Care Research (NIHR). The salary for HS was funded by the NIHR Health Services and Delivery Research programme as part of the PARITY study (NIHR134282).”

“SS is an Academic Clinical Fellow whose salary is part funded by the National Institute for Health and Care Research (NIHR). The salary for HS was funded by the NIHR Health Services and Delivery Research programme as part of the PARITY study (NIHR134282). The views expressed are those of the authors and not necessarily those of the NIHR or the Department of Health and Social Care. The funder did not play any role in the study design, data collection and analysis, decision to publish, or preparation of the manuscript.”

Reviewers' comments:

Reviewer's Responses to Questions

**Comments to the Author**

1. Is the manuscript technically sound, and do the data support the conclusions?

Reviewer #1: Yes

Reviewer #2: Partly

2. Has the statistical analysis been performed appropriately and rigorously?

Reviewer #1: N/A

Reviewer #2: I Don't Know

3. Have the authors made all data underlying the findings in their manuscript fully available?

Reviewer #1: No

Reviewer #2: No

4. Is the manuscript presented in an intelligible fashion and written in standard English?

Reviewer #1: Yes

Reviewer #2: Yes

Reviewer #1: This (?) systematic review has sought to understand the evidence base on patient facing resources on prehabilitation and determine whether motivation and accessibility for patients was balanced. They studied 68 different resources and found moung other things that overpromoting the benefits of prehabilitation may risk giving patients unrealistic expectations. On the whole, the paper is well written and represents a significant addition to the literature on the subject. I have the following comments.

1. Abstract: “Databases, search engines and websites (identified by experts)”…. Who is an expert?

2. This appears to be a systematic review, but the authors have not provided any evidence of prospective registration of the protocol. Hence, it is not possible to decide if the students followed the predetermined protocol. This makes it PRISMA non-compliant. Please clarify.

3. The Introduction runs into 4 pages and should be made more succinct. On the whole, the paper is very long and could be abridged.

4. Why did the authors limit their search to UK data? This limits generalisability.

5. There is no data-sharing statement.

6. All included studies/resources have not been referred to in the bibliography

Reviewer #2: I leave it up to the editor to verify if all the data has been available. The study seems to be well planned and adds evidence to the area of Prehab in cancer rehabilitation. The editor needs to verify if the statistical part is correct since that is not my filed of expertise. This study is also replicable in other geographical locations

**Do you want your identity to be public for this peer review?** For information about this choice, including consent withdrawal, please see our Privacy Policy

Reviewer #1: No

Reviewer #2: No

---

## [Author Response · Author response to Decision Letter 1]

5 Nov 2025

Please find our response to the comments raised by the reviewers and editor in our Response to Reviewers document.

---

## [Decision Letter · Decision Letter 1]

2 Dec 2025

Promoting or pressurising participation? A discourse analysis of online patient information resources about prehabilitation before cancer treatment

PONE-D-25-37313R1

Dear Dr. Stanley,

We’re pleased to inform you that your manuscript has been judged scientifically suitable for publication and will be formally accepted for publication once it meets all outstanding technical requirements.

Kind regards,

Prateek Srivastav

Academic Editor

PLOS ONE

Additional Editor Comments (optional):

Reviewers' comments:

Reviewer's Responses to Questions

**Comments to the Author**

Reviewer #1: All comments have been addressed

2. Is the manuscript technically sound, and do the data support the conclusions?

Reviewer #1: Yes

3. Has the statistical analysis been performed appropriately and rigorously?

Reviewer #1: N/A

4. Have the authors made all data underlying the findings in their manuscript fully available?

Reviewer #1: No

5. Is the manuscript presented in an intelligible fashion and written in standard English?

Reviewer #1: Yes

Reviewer #1: Thank you for responding to my comments and clarifying the points. The revisions to the manuscript are appropriate.

**Do you want your identity to be public for this peer review?** For information about this choice, including consent withdrawal, please see our Privacy Policy

Reviewer #1: No

---

## [Editor Report · Acceptance letter]

PONE-D-25-37313R1

PLOS One

Dear Dr. Stanley,

I'm pleased to inform you that your manuscript has been deemed suitable for publication in PLOS One. Congratulations! Your manuscript is now being handed over to our production team.

Kind regards,

on behalf of

Dr. Prateek Srivastav

Academic Editor

PLOS One